# GerPaCyst - The trial protocol of the prospective, multicenter, interdisciplinary German Pancreas Club Cyst Registry

**Kim Christin Honselmann**[1], **Jonathan Marschner**[1], **Anna Staufenbiel**[1], **Julia Bertram**[1], **Steffen Deichmann**[1], **Carsten Engelke**[2], **Martha Kirstein**[2], **Jens Marquardt**[2], **Marko Damm**[3], **Fanny Borowitzka**[4], **Veit Phillip**[5], **Ilaria Pergolini**[6], **Felix Harder**[7], **Rickmer Braren**[7], **Timo Gemoll**[1], **Alexander Kleger**[8,9], **Matthaeus Felsenstein**[10], **Sophie Schlosser-Hupf**[11], **Matthias Birth**[12], **Louisa Bolm**[1], **Ruediger Braun**[1], **Ihsan Ekin Demir**[6], **Maximilian Denzinger**[8], **Beate Drews**[12], **Ana Dugic**[13], **Thomas Ewers**[2], **Josephine-Alisa Grandi**[10], **Susanne Haberer**[12], **Christoph Ammer-Hermenau**[14], **Philipp Hildebrandt**[15], **Felix Huettner**[16], **Katja Janke**[15], **Katja Kilani**[8], **Georg Lamprecht**[4], **Jolina Michaelis**[1], **Albrecht Neesse**[14], **Lukas Perkhofer**[8,9], **Sebastian Rasch**[5], **Maximilian Reichert**[5], **Thorben Sauer**[1], **Martina Mueller-Schilling**[11], **Jens Schuette**[1], **Rene Wilke**[3], **Susanne Roth**[13], **Sebastian Krug**[17], **Christoph W. Michalski**[13], **Robert Jaster**[4], **Tobias Keck**[1]*, **Ulrich Friedrich Wellner**[1], on behalf of the GerPaCyst Study Group¶

**1** Department of Surgery, University Medical Center Schleswig-Holstein, Campus Luebeck, Lübeck, Germany, **2** Department of Internal Medicine I, University Medical Center Schleswig-Holstein, Campus Luebeck, Lübeck, Germany, **3** Department of Internal Medicine I, University Medical Center Halle, Halle, Germany, **4** Department of Internal Medicine, University Medical Center Rostock, Rostock, Germany, **5** TUM School of Medicine and Health, Department of Clinical Medicine – Clinical Department for Internal Medicine II, University Medical Center, Technical University of Munich, Munich, Germany, **6** Department of Surgery, University Medical Center, Technical University of Munich, Munich, Germany, **7** Department of Radiology, University Medical Center, Technical University of Munich, Munich, Germany, **8** Section of Interdisciplinary Pancreatology, Department of Internal Medicine I, Ulm University Hospital, Ulm, Germany, **9** Institute of Molecular Oncology and Stem Cell Biology, Ulm University Hospital, Ulm, Germany, **10** Department of Surgery, Charité University Medical Center Berlin, Berlin, Germany, **11** Department of Internal Medicine I, University Medical Center Regensburg, Regensburg, Germany, **12** Department of Surgery, Helios Hanseatic Hospital Stralsund, Stralsund, Germany, **13** Department of Gastrointestinal & Infectious Diseases and Surgery, University Medical Center Heidelberg, Heidelberg, Germany, **14** Department of Gastroenterology, Gastrointestinal Oncology and Endocrinology, University Medical Center Goettingen, Goettingen, Germany, **15** Department of Surgery, Hospital Neustadt, Neustadt/Holstein, Germany, **16** Department of Surgery, University Hospital Ulm, Ulm, Germany, **17** Department of Gastroenterology, University Medical Center Heidelberg, Heidelberg, Germany

¶ A full membership list of the GerPaCyst Study Group is provided in the Acknowledgements.
* tobias.keck@uksh.de

## Abstract

### Introduction

Cystic lesions of the pancreas have continued to present a clinical challenge for the past decades now. The increasing rate of detection, the lack of high-quality data on the natural biology of pancreatic cysts and the resulting difficulty to predict malignant transformation in different types of pancreatic cysts make patients with these diseases hard to manage. The German Pancreas Club Cyst Registry (GerPaCyst) (DRKS00025927) aims to establish a platform to discover and survey the natural and

**Data availability statement:** No datasets were generated or analysed during the current study. All relevant data from this study will be made available upon study completion.

**Funding:** Parts of the study were funded by the Federal Ministry of Education and Research (BMBF, 01KD2412A) and further supported by the Clinician-Scientist School Luebeck (project number 413535489) to KCH. The funders had no role in study design, data collection and analysis, decision to publish, or preparation of the manuscript.

**Competing interests:** The authors have declared that no competing interests exist.

**Abbreviations:** IPMN, intraductal papillary-mucinous neoplasia; MCN, mucinous-cystic neoplasia; SCA, serous cystic adenoma; SPN, solid pseudopapillary neoplasia; SPPT, solid pseudopapillary tumors.

specific biology of pancreatic cysts such as IPMNs (main and branch ducts), SCNs, SPPT and MCNs, in a multicenter manner.

## Materials and methods

This manuscript is written according to the SPIRIT guidelines (See S2 and S3 Tables). Ethical approval was obtained from the University of Luebeck (2020-20-225) and all participating centers. In GerPaCyst patients aged ≥18 years with a pancreatic cyst under surveillance or scheduled for surgery should be enrolled. Participating centers will complete an electronic Case Report Form (eCRF) via REDCap which is designed as a longitudinal study minimizing the input of repeated measures. Changes in patient baseline data, cyst characteristics, both endoscopic and imaging data will be entered typically every 6–12 months during patient follow-up. Biobanking will be performed, when available. Duration of observation per patient is up to a maximum of 20 years or until end of follow-up or death.

## Results

The primary goal is to assess and calculate individual risk models for malignancy/high-grade dysplasia based on the collected clinical, molecular and imaging data (multi-omics prediction models) for each included cyst entity, therefore the primary outcome of this trial is the development of high-grade dysplasia/invasive cancer during follow-up or the absence of it. The secondary outcomes are death, quality of life measured by EQ-5D-5L questionnaire, end-of-follow up and perioperative characteristics, if applicable such as complications, length of hospital stay and clavien-dindo classification. Another goal will be to build a multicenter, interdisciplinary database to generate high-quality cyst biology data, which can then be used for further research questions. Additionally, this database will be utilized for registry- based interventional trials in the future.

## Discussion

GerPaCyst will provide a valuable platform for clinical outcomes research. Fundamental factors affecting the development of pancreatic cysts over time will be identified. New research questions might be answered during the study period and will be made available through continuous publications.

## Trial registration

The study was prospectively registered at the German Clinical Trial Register (DRKS) under DRKS00025927 on September 14th, 2021, before inclusion of the first patient. The Universal Trial Number (UTN) is U1111-1302–9822.

## Introduction

Cystic lesions of the pancreas can either be neoplastic or non-neoplastic. Neoplastic cystic lesions include intraductal mucinous papillary neoplasms (IPMN), mucinous

cystic neoplasms (MCN), serous cystic neoplasms (SCN) and solid pseudopapillary tumors (SPPT) and make up 90% of all neoplastic cystic tumors of the pancreas [1]. Depending on the entity, pancreatic cysts bear malignant potential that is associated with outcomes comparable to pancreatic cancer [2].

However, diagnostic accuracy remains limited [3]. The accuracy of cross-sectional imaging varies between 39.5% and 46%.

Pancreatic cysts occur in up to 45% in the German population [4]. The incidence is roughly 13% in Germany and the management of pancreatic cysts has therefore become a highly relevant clinical issue. Recent improvements in high-resolution imaging that make their detection more frequent stands in contrast to treatment algorithms that remain highly controversial in the absence of high- quality controlled studies. Between 2012 and 2018 international guidelines for the treatment of pancreas cysts were published [5–8]. All these guidelines make different recommendations on surveillance of small cysts after five years [9].

However, most of these recommendations are under the bias of retrospective surgical series and observational studies [10,11]. Thus, they probably overestimate the potential for malignancy. Analyses that go beyond five or even ten years hardly exist, which makes it difficult to predict the biology of these entities [12]. Furthermore, the knowledge about the long- term risk of IPMN developing in pancreatic cancer is contradictory. Crippa et al. reported an overall risk of 3.7% of IPMNs to develop into pancreatic cancer [13]. Choi et al. reported that a low risk IPMN (lesion without main pancreatic duct involvement or mural nodes) has a 7.8% chance after 10 years to progress into pancreatic cancer and a high-risk IPMN (mural nodes and dilatated main pancreatic duct) has 25% chance after 10 years to progress into pancreatic cancer [14].

In essence, the natural biologic history of most of the pancreatic cysts we treat today, is unclear. Therefore, we aim to establish the GERPACYST registry, a prospective, 20-year spanning, interdisciplinary, multicenter registry to structurally study the natural biology of different pancreatic cysts over time in order to build multidimensional models to calculate the individual's risk for the development of malignancy.

## Materials and methods

### Aims and scope of the registry study

The registry was developed to provide a framework for studying the natural history of pancreatic cystic lesions. The study idea was initially established at the German Pancreas Club meeting 2020 and is endorsed and sponsored by the German Pancreas Club (DPC). Regarding the above-mentioned aim, the following aspects were especially considered by the founding team (KCH, JM, CWM, SK, RJ, PV, MD, TK, PV, SR and UFW). Relevant outcomes and patient variables were chosen by the founding team. Definitions of the International Study Group of Pancreatic Surgery (ISGPS), the TNM classification system, the German S3 guideline for the treatment of pancreatic cancer and the Fukuoka guidelines were utilized. The updated Kyoto guidelines were added August 13th, 2024. Ethical approval was obtained from the University of Luebeck (2020-20-225)/(09/07/2020), first patient entered:14/01/2022. Participating centers have obtained approval from their respective ethics commission prior to data collection: University Medical Center Rostock (18/10/2021, first patient entered: 16/02/2023, University Medical Center Goettingen (25/10/2021, first patient entered: 24/01/2023), Technical University of Munich (26/10/2021, first patient entered: 11/07/2023), University Medical Center Halle (31/01/2022, first patient entered 23/04/2025), Ulm University Hospital (29/08/2022, first patient entered: 22/04/2024), University Medical Center Heidelberg (26/01/2024, first patient entered: 12/07/2024), Helios Hanseatic Hospital Stralsund (06/09/2024, first patient entered: 14/03/2025). The requirement for separate ethics approvals from Charité University Medical Center Berlin, University Medical Center Regensburg and Schoen Hospital Neustadt was waived by the ethics committees of the three participating hospitals, on the basis that their review processes recognize and accept the existing ethics approval granted by the University of Luebeck (2020-20-225). University Medical Center Regensburg signed their waiver on 30/09/2021 and included their first patient 28/04/2025. Charité University Medical Center Berlin became a part of the study on 04/12/2024 and Schoen Hospital Neustadt joined on 07/09/2022. Their first patient was included on the 09/12/2024 (Berlin) and on the

25/10/2022 (Neustadt), respectively. Patients participating in the study are informed beforehand and give written consent for their data to be collected. The end of the study after its 20-year period is the 31/09/2041 for all centers.

## Study design

The GerPaCyst Study is a prospective, multicenter, interdisciplinary, longitudinal registry study. A formal protocol detailing aims and regulations for data safety and publication has been uploaded to https://drks.de/search/en/trial/DRKS00025927 and can be accessed publicly. This manuscript also provides a concise form to fulfill standards for documentation of registries and follows the Standard protocol items: recommendations for clinical trials (SPIRIT) guidelines [15].

## Registry organization

The GerPaCyst-registry is headed by the Steering Committee including KCH, MD, RJ, SK, CM and UFW. The registry is physically located at the University of Luebeck, Germany. Maintenance is secured by the IT-Service Center Luebeck.

## Participants, eligibility criteria and outcomes

Patients presenting at the respective participating centers including academic and academic-affiliated hospitals, with a presumed or confirmed pancreatic cyst (IPMN, MCN, SPPT, or SCN) will be evaluated for participating in the study. The diagnosis of a presumed pancreatic cysts will be established by the treating physician in accordance with the European evidence-based guidelines on pancreatic cystic neoplasms, the international, Sendai, Fukuoka criteria and the updated Kyoto guidelines [5,16–18], which defines the clinical, imaging and laboratory features required for case inclusion. Diagnosis criteria include the following:

- Main-Duct IPMN is suspected for segmental or diffuse dilation of the main pancreatic duct (MPD) of >5 mm without other causes of obstruction.

- Any cyst above or equal 1 cm should be checked by CT or MRI with MRCP to check for high-risk stigmata, including enhanced solid component and main pancreatic duct size of more than 1 cm, or worrisome features", including cyst over or equal 3 cm, thickened enhanced cyst walls, non-enhanced mural nodules, MPD size 5–9 mm, abrupt change in MPD caliber with distal pancreatic atrophy and lymphadenopathy

- All cysts over 3 cms or with worrisome features or high-risk stigmata" should undergo EUS. MDCT and MRCP are most useful for distinguishing BD-IPMN from other cysts by showing multiplicity and a connection to the MPD. Cyst fluid is optional but can help to distinguish a BD-IPMN from a small oligocystic SCN with CEA determination.

- The distinction between main, mixed or branch-duct type IPMN should be made upon imaging. The pathological classification can only be done postoperatively.

- Neoplastic transformation will be separated into low-grade and high-grade dysplasia, as well as invasive cancer.

- MCN should be restricted to neoplasms exhibiting ovarian-type stroma.

- SPN/SPPTs are well-circumscribed mass with calcification, peripheral capsule, internal blood products, and lack of biliary/pancreatic ductal obstruction) on computed tomography and MRI are highly suggestive of the diagnosis of SPN, particularly when visualized in young female patients.

   Imaging will be subjected to review at the University Medical Center Schleswig-Holstein, Campus Luebeck upon study inclusion for the prospective risk stratification model. For future research questions, MR and CT images will be reviewed at random to ensure quality of entered data. Discrepancies will be recorded and used to calculate inter-observer variability.

**These are the inclusion and exclusion criteria:**

**Inclusion criteria:**

- Presumed pancreatic cyst of the following entities (IPMN, MCN, SPPT, SCN)

- At least 18 years old

**Exclusion criteria:**

- Confirmed pseudocyst or diagnosis other than IPMN, MCN, SPPT, SCN

- Less than 18 years old

Suitable patients will be informed about the research project by the treating physician. The written informed consent will be obtained by each patient. The treatment interventions will not be altered by the study.

If the patient provides consent to participate in GerPaCyst, data entry will take place at the time of treatment initiation (surgery initiation of non-surgical surveillance) and every 6–12 months thereafter, for a period of up to 20 years. Participating centers complete an electronic Case Report Form (eCRF). Study data is collected and managed using REDCap electronic data capture tools hosted at the University of Luebeck, Germany [19,20] (Variables see S1 Table). Changes in patient characteristics are entered during patient follow-up. In addition, patient biobank samples (blood, cyst fluid, urine, tissue) will be stored upon availability.

The primary outcome of this trial is the development of high-grade dysplasia/invasive cancer during follow-up or the absence of it. The secondary outcomes are death, quality of life measured by EQ-5D-5L questionnaire, end-of-follow up and perioperative characteristics if applicable such as complications, length of hospital stay and clavien-dindo classification.

### Initiation algorithm

The study is still welcoming new participating centers from all over the world. Interested centers can contact: kimchristin.honselmann@uksh.de or redcap.lachi@uni-luebeck.de. Once the study protocol has been reviewed by the ethics committee of the new center, and the contract has been approved by both legal departments, the participating physicians will receive proper training for patient inclusion in the REDCap based database.

### Participant timelines and follow-up schedules

Only after the patient has given written informed consent may the treating physician register the patient in the database. After patient registration, mandatory baseline data on premedical history, data on the disease, imaging and diagnostic modalities, treatment, and treatment outcome (e.g., post-op complications) are documented. In addition, the date of treatment initiation, patient number, and basic patient-specific information are documented (e.g., gender or month and year of birth). The study's non-interventional design does not regulate modalities or intervals between examinations and thus does not intervene in routine clinical practice. Patients will be followed when presenting to the participating institutions or by telephone interview every 6–12 months. A detailed patient schedule is shown (Fig 1).

### Patient rights and data safety

Written informed consent for participation in the registry is obtained prior to inclusion at every center from all patients. Patient data are entered and stored in pseudonymized form. Institutional review boards have been obtained from every participating center. A data safety manual can be obtained from the senior author (KCH). The GerPaCyst platform was initiated and enabled for data collection in pancreatic cysts in 2021 (Ethics Committee of the University of Lübeck Az. 20−225 and 2024–265_1).

| | STUDY PERIOD | | | | | | | |
|---|---|---|---|---|---|---|---|---|
| | Enrolment | Baseline | Follow-up | | | | | Close out |
| **TIMEPOINT\*\*** | **-t₁** | **0** | **t₁** | **t₂** | **t₃** | **t₄** | **etc.** | **tₓ** |
| **ENROLMENT:** | | | | | | | | |
| **Eligibility screen** | X | | | | | | | |
| **Informed consent** | X | | | | | | | |
| ***EQ-5D-5L*** | X | | | | | | | x |
| **Blood sample for biobanking** | | X | x | x | x | x | x | x |
| **Standard-of- care:** | | | | | | | | |
| ***Imaging*** | | x | x | x | x | x | x | |
| ***Surgery*** | | ? | ? | ? | ? | ? | ? | |
| **ASSESSMENTS:** | | | | | | | | |
| ***Baseline variables (Age, sex, date of first visit)*** | | X | x | x | x | x | x | |
| ***Comorbidities and symptoms*** | | x | % | % | % | % | % | % |
| ***Cyst variables (Size, number, characteristics)*** | | x | X | X | X | X | x | X |

% Update of any new comorbidities or symptoms that have developed during follow-up
?: Surgery might occur at baseline or during follow-up

**Fig 1. SPIRIT schedule of enrollment, interventions, and assessment.**

## Assessment of data

After browser-based data entry, the data are automatically checked for completeness and internal plausibility by software-based internal rules. Missing or implausible items create warning messages. The data is usually entered after seeing the patient in the outpatient clinic of the respective institution. Missing data will be handled according to the intended analysis. There is currently no external monitoring for the GerPaCyst Registry. Data are collected prospectively via standardized physician-completed questionnaires. Diagnosis will be made according to the standardized criteria outlines in the aforementioned guidelines. Regular quality control is performed by semiannual study meetings of the entire study group. Through these periodic audits harmonized data collection is ensured, combined with the use of the electronic database that automatically identifies and flags incomplete case forms. The GerPaCyst steering committee recognizes that some methodological aspects may require refinement during implementation. Any such changes will be documented and reported. Specifically, laboratory analyses and parameters used in risk models, as well as imaging parameters, may be subject to change over the course of the study.

For building a multi-omics risk stratification model for pancreatic cyst progression, we will utilize deep learning methods. First, we will ensure data processing and integration and preprocess data by normalization (proteomics, genomics, metabolomics, etc.) and harmonization. Data will further be integrated by feature selection and dimensions will be reduced.

Then deep learning will be integrated such as neuronal networks. These calculations will be performed by designated bioinformaticians.

## Database structure

GerPaCyst is based on a web-based REDCap data entry system for the systematic collection of patient data, which was developed by Vanderbilt University [19]. Data entry is paperless and performed directly on-site at the respective medical facility (clinic/practice) via an Internet browser. The REDCap implementation meets all requirements of Good Clinical Practice (ICH-GCP) as well as European data safety regulations. Corresponding certificates for data security are available. The system is operated on a server of the computer center of the University of Lübeck and is thus subject to strict access control. Data is collected in pseudonymized form. Pseudonymization keys are kept separately from the database at the participating institutions.

Duration of observation per patient is up to a maximum of 20 years or until the occurrence of an event defined as end of follow-up or death. The study start was the fourth quarter of 2021 and is still ongoing. Centers are still being initialized at the moment.

Entered variables include baseline characteristics such as data on comorbidities like arterial hypertension, diabetes mellitus, chronic obstructive lung disease, medical history of cancer, family history and past medical history. Also, data on perioperative characteristics, if applicable, will be collected. Imaging modalities on the most prominent cysts will be collected at each follow-up. We will perform a quality-of-life questionnaire (EQ-5D-5L) at enrollment and at end of follow-up or surgery [21]. Cyst fluid analysis/molecular characteristics will be collected, if available (S1 Table).

## Statistics

The data required for the respective research question are compiled into an anonymized project-specific evaluation data set with special consideration of the resulting re-identification risk, if any, and made available for the research project. The GerPaCyst study group decides on the provision of the evaluation data sets for scientific evaluation.

For the development of risk models, we aim to utilize neuronal networks with elastic-net regression using R. ROC-Curves will be built for candidate risk variables. According to our and the experience of others, sample size should be ten to 20 times higher than the number of trainable parameters. As of now, we are planning to include 157 variables plus imaging and biomarker profiles (about 40 different parameters). Our sample size should therefore be at least 4000 patients (200x20). To achieve adequate participant enrolment, this study was set up as an international multicenter study. Descriptive data will be evaluated with R or SPSS according to the data characteristics. Nominal data will be evaluated with frequencies and percentages and differences with Chi-square test or the Kaplan-Meyer methods if pertaining to survival. Continuous data will be evaluated by median and interquartile range, differences between two groups will be calculated by Mann-Whitney U Test. Kaplan-Meyer approximation will be used for survival analyses. P-value will be set at $p < 0.05$.

The further exact analyses will be defined in a Statistical Analysis Plan (SAP) prior to the corresponding evaluation. However, to ensure appropriate analyses, the nature of the data must be considered. For this reason, descriptive analyses will be conducted, from which new hypotheses and questions will arise. Potential systematic biases must be identified and accounted for in the analyses. To account for interobserver variability in radiological data sets, each time a scientific question is raised, it is followed by a review of the imaging data supplied. Other important decisions include defining subgroups of interest (e.g., patient groups that are usually excluded from RCTs) or pooling similar therapies (e.g., when there are only a few patients in individual therapies) and dealing with missing values in registry data. Some of the questions result in periodic evaluations. Other questions arise due to the ongoing, observational process only during the study, which then result in supplementary SAPs. Answering questions that arise only in the course of the study is one of the great advantages (and difficulties) of registry studies [22].

Only the ITSC of the University of Lübeck and the leaders of the GerPaCyst study group or persons appointed by the study group, such as staff of the Chir-NET office of the University Hospital Schleswig-Holstein, have access to the complete data set.

The GerPaCyst registry has been established to be used together with the StuDoQIPancreas Registry, if necessary and for facilitating data use and transfer within Germany, but also internationally. Accordingly, many items have been adapted from the StuDoQIPancreas registry [23].

### Dissemination policy

Trial results will be communicated with all participating centers on a regular basis including half-yearly meeting sessions at the German Pancreas Club and the German Visceral Medicine meetings (AG Pankreas DGVS). In addition, 4-monthly group meetings will be held virtually in order to communicate possible protocol changes or interim-results. Publication will be decided upon by the steering committee and author inclusion will be based on patient inclusion: up to 10 patients per year: 1 author from each center, 10–20 patients per year: 2 authors per center, etc. Requests for data analysis of the whole cohort can be send to the steering committee for evaluation. The full protocol has been uploaded to the German clinical trial registry (www.drks.de). Participant-level dataset and statistical codes can be obtained from KCH.

### Biological specimens

Blood samples or cyst (fluid) biopsies or surgical specimens will be collected at the participating centers, if available and will possibly be used for genetic and molecular analysis in ancillary studies in the future.

### Appendices

Informed consent (Currently in German).

### Discussion

More than twenty years after the first introduction of intraductal papillary mucinous neoplasms (IPMN) by the World Health Organization (WHO), we still struggle to treat not only IPMNs, but most pancreatic cysts. The increasing frequency of detection caused by the rising number of radiology imaging exams will make this an even more pressing issue in the not so far future.

A large surgical series of 577 branch-duct IPMNs (BD-IPMN) found that at a median follow-up of nearly seven years, almost 50% of BD-IPMNs progressed over time with 34% of patients having either worrisome features or high-risk stigmata at the end of follow up [10]. In contrast, a radiologic group found no development of malignancy in a group of 49 patients with BD-IPMNs less than 2 cm in diameter [24]. Similarly, in an Asian study only 9% of 722 patients developed malignancy over a five-year follow up-period [25]. A recent large study from Verona showed that the risk of developing malignancy in presumed BD-IPMN, that are stable over five years and less than 30 mm in size was comparable to that of the general population in the same age group. They even proposed to discontinue surveillance in patients older than 75 years [26]. However, the median follow-up was less than five years in this study. Furthermore, the discrepancies between these studies might be attributed to the different sources of patients (from surgery, radiology and gastroenterology separately).

Another point of discussion in the treatment of pancreatic cysts are main-duct IPMNs and the questions if we should resect all of them, even in old and frail patients. Crippa et al. found that the impact on survival was low, even in those IPMNs with a high malignancy rate [27]. Mucinous cystic neoplasms are to be resected, if they are larger than four centimeters [28]. But MCNs under four centimeters also had a malignancy rate of 43% in a retrospective study by Hohn et al. [29]. These recommendations are also based on retrospective surgical series, which might overestimate the rate of malignancy.

Based on data of pancreatic cancer specimens, clonal evolution from the initiation of tumorigenesis until the birth of the cell giving rise to the parental clone of primary cancer takes an estimate of 11.7 years, an average of 6.8 years from then until the birth of the cell giving rise to the index primary lesion, and an average of 2.7 years from then until the patients' death [30]. In total, the time of tumorigenesis, here in the example of pancreatic cancer, takes 20.2 years until metastatic dissemination and death. Maybe, we can assume that in the case of pancreatic cysts, the timing is similar or even slower, illustrating the need for long-term follow-up data.

The GerPaCyst registry is therefore designed to include both patients treated by gastroenterologists, radiologist and surgeons to reduce the bias of either surveillance or resected patients. The web-based format and longitudinal study set up is supposed to facilitate inclusion and follow-up and the duration of 20 years will hopefully give the opportunity to include a large group of patients.

With artificial intelligence (AI) on the rise, GerPaCyst will provide a well characterized cohort to perform AI analyses with and to bring pancreatic cyst risk calculation to the next scientific level.

In conclusion, we established a prospective, multicenter, interdisciplinary pancreatic cyst registry to enable the uncovering of long-term pancreatic cyst biology and hope to contribute to improved patient care in the future.

## Trial status

11 university hospitals throughout Germany are currently participating in this study. The first patient was entered in December 2021 and the study is ongoing. It is an interdisciplinary cooperation of surgeons. radiologists and gastroenterologists. At least 5000 patients are planned to be recruited from gastroenterology and surgical departments. Baseline characteristics such as metabolic/endocrine diseases, risk factors and family history, as well as pancreatic cyst data including MPD diameter, enhanced mural nodules, calcifications and the quality-of-life EQ-5 questionnaire will be assessed during index visit and follow-up. Biospecimens like blood, cyst fluid and cyst cytology will be collected, if needed.

## Supporting information

**S1 Table. GerPaCyst REDCap variables.**
(DOCX)

**S2 Table. Spirit checklist guidelines.**
(DOCX)

**S3 Table. Human subjects research checklist.**
(DOCX)

**S4 Table. Blood and and imaging parameters.**
(DOCX)

## Acknowledgments

The study is registered at https://drks.de/search/en/trial/DRKS00025927. The protocol has been reviewed and approved by the committees of each participating center.

We thank the IT-Service Center University of Luebeck, especially Maike Wolf and Helge Illig for the establishment and maintenance of the REDCAP Database.

We thank the entire GerPaCyst Study Group for their efforts and their collaborative spirit. All listed authors in this manuscript are part of the GerPaCyst Study Group, lead author is Kim Christin Honselmann (kimchristin.honselmann@uksh.de). The GerPaCyst Study Group are:

Kim Christin Honselmann, University Medical Center Schleswig-Holstein, Lübeck

Jonathan Marschner, University Medical Center Schleswig-Holstein, Lübeck
Anna Staufenbiel, University Medical Center Schleswig-Holstein, Lübeck
Julia Bertram, University Medical Center Schleswig-Holstein, Lübeck
Steffen Deichmann, University Medical Center Schleswig-Holstein, Lübeck
Carsten Engelke, University Medical Center Schleswig-Holstein, Lübeck
Martha Kirstein, University Medical Center Schleswig-Holstein, Lübeck
Jens Marquardt, University Medical Center Schleswig-Holstein, Lübeck
Marko Damm, University Medical Center Halle, Halle
Fanny Borowitzka, University Medical Center Rostock, Rostock
Veit Phillip, Technical University Munich, Munich
Ilaria Pergolini, Technical University Munich, Munich
Felix Harder, Technical University Munich, Munich
Rickmer Braren, Technical University Munich, Munich
Timo Gemoll, University Medical Center Schleswig-Holstein, Lübeck
Alexander Kleger, University Medical Center Ulm, Ulm
Matthaeus Felsenstein, University Medical Center Charite, Berlin
Sophie Schlosser-Hupf, University Medical Center Regensburg, Regensburg
Matthias Birth, Hospital Stralsund, Stralsund
Louisa Bolm, University Medical Center Schleswig-Holstein, Lübeck
Ruediger Braun, University Medical Center Schleswig-Holstein, Lübeck
Ihsan Ekin Demir, Technical University Munich, Munich
Maximilian Denzinger, University Medical Center Ulm, Ulm
Beate Drews, Hospital Stralsund, Stralsund
Ana Dugic, University Medical Center Heidelberg, Heidelberg
Thomas Ewers, University Medical Center Schleswig-Holstein, Lübeck
Josephine-Alisa Grandi, University Medical Center Charite, Berlin
Susanne Haberer, Hospital Stralsund, Stralsund
Christoph Ammer-Hermenau, University Medical Center Göttingen, Göttingen
Philipp Hildebrandt, Hospital Neustadt, Neustadt
Felix Huettner, University Medical Center Ulm, Ulm
Katja Janke, Hospital Neustadt, Neustadt
Katja Kilani, University Medical Center Ulm, Ulm
Georg Lamprecht, University Medical Center Rostock, Rostock
Jolina Michaelis, University Medical Center Schleswig-Holstein, Lübeck
Albrecht Neesse, University Medical Center Schleswig-Holstein, Lübeck
Lukas Perkhofer, University Medical Center Ulm, Ulm
Sebastian Rasch, Technical University Munich, Munich
Maximilian Reichert, Technical University Munich, Munich
Thorben Sauer, University Medical Center Schleswig-Holstein, Lübeck
Martina Mueller-Schilling, University Medical Center Regensburg, Regensburg
Jens Schuette, University Medical Center Schleswig-Holstein, Lübeck
Rene Wilke, University Medical Center Halle, Halle
Susanne Roth, University Medical Center Heidelberg, Heidelberg
Sebastian Krug, University Medical Center Halle, Halle

Christoph W. Michalski, University Medical Center Heidelberg, Heidelberg

Robert Jaster, University Medical Center Rostock, Rostock

Tobias Keck, University Medical Center Schleswig-Holstein, Lübeck

Ulrich Friedrich Wellner, University Medical Center Schleswig-Holstein, Lübeck

Patients are informed beforehand and give consent to participate in written form.

## Author contributions

**Conceptualization:** Kim Christin Honselmann, Marko Damm, Fanny Borowitzka, Veit Phillip, Felix Harder, Rickmer Braren, Robert Jaster.

**Data curation:** Kim Christin Honselmann, Jonathan Marschner, Anna Staufenbiel, Julia Bertram, Ilaria Pergolini, Sophie Schlosser-Hupf, Jens Schuette, Martina Mueller-Schilling.

**Funding acquisition:** Kim Christin Honselmann, Timo Gemoll.

**Investigation:** Marko Damm, Fanny Borowitzka, Veit Phillip, Ilaria Pergolini, Felix Harder, Rickmer Braren, Timo Gemoll, Alexander Kleger, Matthaeus Felsenstein, Sophie Schlosser-Hupf, Susanne Roth, Sebastian Krug, Christoph W. Michalski, Robert Jaster, Matthias Birth, Louisa Bolm, Ruediger Braun, Ihsan Ekin Demir, Maximilian Denzinger, Beate Drews, Ana Dugic, Thomas Ewers, Josephine-Alisa Grandi, Susanne Haberer, Christoph Ammer-Hermenau, Philipp Hildebrandt, Felix Huettner, Katja Janke, Katja Kilani, Georg Lamprecht, Jens Schuette, Jolina Michaelis, Albrecht Neesse, Rene Wilke, Lukas Perkhofer, Sebastian Rasch, Maximilian Reichert, Martina Mueller-Schilling, Thorben Sauer, Tobias Keck, Ulrich Friedrich Wellner.

**Methodology:** Kim Christin Honselmann, Jonathan Marschner, Anna Staufenbiel, Carsten Engelke, Matthaeus Felsenstein, Robert Jaster, Beate Drews, Ana Dugic, Sebastian Rasch.

**Project administration:** Kim Christin Honselmann, Jonathan Marschner, Anna Staufenbiel, Julia Bertram, Jolina Michaelis.

**Resources:** Julia Bertram, Steffen Deichmann, Carsten Engelke, Martha Kirstein, Jens Marquardt, Alexander Kleger, Matthaeus Felsenstein, Maximilian Denzinger, Georg Lamprecht, Maximilian Reichert.

**Software:** Thorben Sauer.

**Supervision:** Rickmer Braren, Sebastian Krug, Christoph W. Michalski, Robert Jaster, Tobias Keck, Ulrich Friedrich Wellner.

**Writing – original draft:** Kim Christin Honselmann.

**Writing – review & editing:** Kim Christin Honselmann, Jonathan Marschner, Anna Staufenbiel, Julia Bertram, Steffen Deichmann, Carsten Engelke, Martha Kirstein, Jens Marquardt, Marko Damm, Fanny Borowitzka, Veit Phillip, Ilaria Pergolini, Felix Harder, Rickmer Braren, Timo Gemoll, Alexander Kleger, Matthaeus Felsenstein, Sophie Schlosser-Hupf, Susanne Roth, Sebastian Krug, Christoph W. Michalski, Robert Jaster, Matthias Birth, Louisa Bolm, Ruediger Braun, Ihsan Ekin Demir, Maximilian Denzinger, Beate Drews, Ana Dugic, Thomas Ewers, Josephine-Alisa Grandi, Susanne Haberer, Christoph Ammer-Hermenau, Philipp Hildebrandt, Felix Huettner, Katja Janke, Katja Kilani, Georg Lamprecht, Jens Schuette, Jolina Michaelis, Albrecht Neesse, Rene Wilke, Lukas Perkhofer, Sebastian Rasch, Maximilian Reichert, Martina Mueller-Schilling, Thorben Sauer, Tobias Keck, Ulrich Friedrich Wellner.

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
