## [Decision Letter · Decision Letter 0]

8 Aug 2025

Dear Dr. Honselmann,

Thank you for submitting your manuscript to PLOS ONE. After careful consideration, we feel that it has merit but does not fully meet PLOS ONE’s publication criteria as it currently stands. Therefore, we invite you to submit a revised version of the manuscript that addresses the points raised during the review process.

We look forward to receiving your revised manuscript.

Kind regards,

Yoshihisa Tsuji

Academic Editor

PLOS ONE

Journal Requirements:

2. Thank you for stating the following financial disclosure: [Parts of the study are currently funded by The Federal Ministry of Education and Research (BMBF) (Funding number: 01KD2412A) and was supported by the Clinician-scientist school Luebeck (project number 413535489) to KCH.]. 

4. Please amend your authorship list in your manuscript file to include authors Dr. Matthias Birth, Louisa Bolm, Ruediger Braun, Ihsan Ekin Demir, Maximilian Denzinger, Beate Drews, Ana Dugic, Thomas Ewers, Josephine-Alisa Grandi, Susanne Haberer, Christoph Ammer-Hermenau, Philip Hildebrandt, Felix Huettner, Katja Janke, Katja Kilani, Georg Lamprecht, Jens Schuette, Jolina Michaelis, Albrecht Neesse, Rene Wilke, Lukas Perkhofer, Sebastian Rasch, Maximilian Reichert, Martina Mueller-Schilling, and Thorben Sauer.

5. One of the noted authors is a group or consortium [the GerPaCyst Study Group]. In addition to naming the author group, please list the individual authors and affiliations within this group in the acknowledgments section of your manuscript. Please also indicate clearly a lead author for this group along with a contact email address.

6. We notice that your supplementary tables are included in the manuscript file. Please remove them and upload them with the file type 'Supporting Information'. Please ensure that each Supporting Information file has a legend listed in the manuscript after the references list.

Reviewers' comments:

Reviewer's Responses to Questions

**Comments to the Author**

1. Does the manuscript provide a valid rationale for the proposed study, with clearly identified and justified research questions?

Reviewer #1: Yes

Reviewer #2: Yes

2. Is the protocol technically sound and planned in a manner that will lead to a meaningful outcome and allow testing the stated hypotheses?

Reviewer #1: Partly

Reviewer #2: Yes

3. Is the methodology feasible and described in sufficient detail to allow the work to be replicable?

Reviewer #1: No

Reviewer #2: Yes

4. Have the authors described where all data underlying the findings will be made available when the study is complete?

Reviewer #1: No

Reviewer #2: Yes

5. Is the manuscript presented in an intelligible fashion and written in standard English?

Reviewer #1: Yes

Reviewer #2: Yes

You may also provide optional suggestions and comments to authors that they might find helpful in planning their study.

Reviewer #1: This is the protocol paper of the prospective, multicenter German Pancreas Club cyst registry.

Minor Comments:

1. Clarification of presumed diagnosis criteria:

The protocol mentions inclusion of patients with a “presumed diagnosis”. However, it is unclear what specific clinical, imaging, or laboratory criteria are used to define this presumed diagnosis. Please clarify the diagnostic criteria and specify how interobserver variation is addressed in such assessments. This information should be explicitly described in the protocol.

2. Handling and evaluation of imaging data:

It is not clear whether the imaging findings are assessed solely based on site-reported entries in the case report forms or whether a central image review is planned. If central review of imaging data is not conducted, this should be explicitly stated. In that case, please describe how consistency across institutions (e.g., interobserver variability) will be managed or minimized. This point should be clearly addressed in the study protocol.

Reviewer #2: Comment

This is a large observational study of the long-term follow-up of pancreatic cystic lesions in Germany. Prospective studies of pancreatic cystic lesions are limited and the findings from this study are expected to be important. Some comments are provided below.

1. Abstract lacks primary endpoints, making it difficult to understand the study summary.

2. Is there a defined objective number of cases?

3. The evaluation items are difficult to understand. For example, the items to be evaluated in blood tests and in imaging tests. Why not list them all in a table?

**Do you want your identity to be public for this peer review?** For information about this choice, including consent withdrawal, please see our Privacy Policy

Reviewer #1: No

Reviewer #2: No

---

## [Author Response · Author response to Decision Letter 1]

3 Oct 2025

Reviewers' comments:

Reviewer's Responses to Questions

Comments to the Author

1. Does the manuscript provide a valid rationale for the proposed study, with clearly identified and justified research questions?

Reviewer #1: Yes

Reviewer #2: Yes

2. Is the protocol technically sound and planned in a manner that will lead to a meaningful outcome and allow testing the stated hypotheses?

Reviewer #1: Partly

Reviewer #2: Yes

Answer 2: For Reviewer #1: Since this is not an interventional trial, controls are not described. We have added methodology for building a multi-omics risk stratification model in line 275ff.

3. Is the methodology feasible and described in sufficient detail to allow the work to be replicable?

Reviewer #1: No

Reviewer #2: Yes

Answer 3: Reviewer#1: We see your concerns. As this is a registry trial, the research questions in the future are not sufficiently foreseeable at this point. However, our primary goal is to generate a multi-omics model for risk stratification of pancreatic cyst progression. For this, we added the detailed methodology and hope to answer your question sufficiently (Line 282ff.

4. Have the authors described where all data underlying the findings will be made available when the study is complete?

The PLOS Data policy requires authors to make all data underlying the findings described in their manuscript fully available without restriction, with rare exception, at the time of publication. The data should be provided as part of the manuscript or its supporting information or deposited to a public repository. For example, in addition to summary statistics, the data points behind means, medians and variance measures should be available. If there are restrictions on publicly sharing data—e.g. participant privacy or use of data from a third party—those must be specified.

Reviewer #1: No

Reviewer #2: Yes

Answer 4: We thank Reviewer #1 and #2 for this comment. As we have described and outlined above, this is a mere study protocol. There is no data available currently as the study is still in its data acquisition phase. There are no concrete data points to be shared at this moment. As our study continues, we will make the outcomes available through publications. We have added a comment regarding publication of our findings in the manuscript line 354f: “Participant-level dataset and statistical codes can be obtained from KCH.”

5. Is the manuscript presented in an intelligible fashion and written in standard English?

Reviewer #1: Yes

Reviewer #2: Yes

Answer 5: We thank Reviewer #1 and #2 for their comment.

6. Review Comments to the Author

Reviewer #1: This is the protocol paper of the prospective, multicenter German Pancreas Club cyst registry.

Minor Comments:

1. Clarification of presumed diagnosis criteria:

The protocol mentions inclusion of patients with a “presumed diagnosis”. However, it is unclear what specific clinical, imaging, or laboratory criteria are used to define this presumed diagnosis. Please clarify the diagnostic criteria and specify how interobserver variation is addressed in such assessments. This information should be explicitly described in the protocol.

Answer 1: We added our criteria for diagnosis of the different pancreatic cysts in the manuscript under “Participants, Eligibility criteria and outcomes” line 199ff.

2. Handling and evaluation of imaging data:

It is not clear whether the imaging findings are assessed solely based on site-reported entries in the case report forms or whether a central image review is planned. If central review of imaging data is not conducted, this should be explicitly stated. In that case, please describe how consistency across institutions (e.g., interobserver variability) will be managed or minimized. This point should be clearly addressed in the study protocol.

Answer 2: We thank Reviewer #1 for this comment. We will perform central image review. We have addressed this in line 205ff.

Reviewer #2: Comment

This is a large observational study of the long-term follow-up of pancreatic cystic lesions in Germany. Prospective studies of pancreatic cystic lesions are limited and the findings from this study are expected to be important. Some comments are provided below.

1. Abstract lacks primary endpoints, making it difficult to understand the study summary.

2. Is there a defined objective number of cases?

3. The evaluation items are difficult to understand. For example, the items to be evaluated in blood tests and in imaging tests. Why not list them all in a table?

Answer: We have addressed 1) in the abstract and have added the necessary information in line 86ff. Regarding 2) we have adjusted the manuscript accordingly in line 434f . Regarding comment 3) we have added a table to the supporting information addressing the items that are being evaluated at the moment. (S4 Table). However, this might change over the course of the 20 year study period.

7. PLOS authors have the option to publish the peer review history of their article (what does this mean?). If published, this will include your full peer review and any attached files.

Answer 7: Yes, please include the peer review.

---

## [Editor Report · Decision Letter 1]

17 Oct 2025

GerPaCyst- The Trial Protocol Of The Prospective, Multicenter, Interdisciplinary German Pancreas Club Cyst Registry

PONE-D-25-34705R1

Dear Dr. Honselmann,

We’re pleased to inform you that your manuscript has been judged scientifically suitable for publication and will be formally accepted for publication once it meets all outstanding technical requirements.

Kind regards,

Yoshihisa Tsuji

Academic Editor

PLOS ONE
---

## [Editor Report · Acceptance letter]

PONE-D-25-34705R1

PLOS ONE

Dear Dr. Honselmann,

I'm pleased to inform you that your manuscript has been deemed suitable for publication in PLOS ONE. Congratulations! Your manuscript is now being handed over to our production team.

Kind regards,

on behalf of

Professor Yoshihisa Tsuji

Academic Editor

PLOS ONE